# Practice for Continuous Effect of Technology Transfer in Bridge Maintenance and Management

**Teruyuki Miyakawa [1], Takafumi Nishikawa [2,*], Shozo Nakamura [2] and Yoshimoto Koyanagi [3]**

[1] Dia Nippon Engineering Consultants Co., Ltd., 300 Kandaneribei, Chiyoda, Tokyo 101-0022, Japan; miyakawa_teruyuki@dcne.co.jp

[2] Graduate School of Engineering, Nagasaki University, 1-14 Bunkyomachi, Nagasaki 852-8521, Japan; shozo@nagasaki-u.ac.jp

[3] Japan International Cooperation Agency, 5-25 Nibancho, Chiyoda, Tokyo 102-8012, Japan; koyanagi.yoshimoto@jica.go.jp

* Correspondence: nishikawa@nagasaki-u.ac.jp

**Abstract:** As part of international cooperation efforts by the Japan International Cooperation Agency, Japan has implemented technical cooperation projects for bridge maintenance and management in several countries. The authors conducted technology transfer for bridge inspection and repair in technical cooperation projects in Egypt and the Philippines. This paper describes the past shortcomings and future support methods required based on these efforts as well as maintenance efforts in Japan. At the organizational level, we examined goals and strategies for fostering "effective ownership" and the nature of the PDCA (Plan-Do-Check-Action) cycle, including effectiveness measurement. At the individual engineer level, we examined the need to accumulate and pass on experience. Through these studies, the sustainable management cycle was examined. As a result, the studies identify the importance of flexibility in project monitoring and revision. It is necessary to determine the evaluation period based on actual achievement rather than time constraints. Besides, it is important to elastically reconsider indicators even at higher levels as well as effective countermeasures.

**Keywords:** maintenance management; technical cooperation; experience engineering

## 1. Introduction

A multilateral development bank is an international organization that helps developing countries reduce poverty and achieve economic and social development in a comprehensive manner through financial aid and intellectual contributions. In this, developed countries and developing countries participate as donors and recipients, respectively. The World Bank Group (established in 1946) provides aid to the entire world through its member organizations, including the International Bank for Reconstruction and Development, International Development Association, International Finance Corporation, and Multilateral Investment Guarantee Agency. Regional development banks (banks involved in financing activities according to the characteristics of each region) include the Asian Development Bank (ADB, established in 1966) and the African Development Bank (AfDB, established in 1964), which provide aid to Asian and African countries, respectively.

International aid from donor countries is provided on the basis of their own aid policies and implementing organizations and is expected to be delivered in cooperation with multilateral development banks. Accumulated experience and existing knowledge of international organizations, as well as their politically neutral advice, are essential; the political dialogue, aid coordination, and co-financing are beneficial [1].

On the other hand, Japan's international cooperation can be divided into paid financial cooperation, grant aid, and technical cooperation [2]. While financial cooperation focuses on constructing and maintaining infrastructure, grant aid is provided for developing hospitals, schools, water supply facilities, human resource development, and other activities.

Technical cooperation includes surveys and plans, the establishment of organizations and institutions, human resource development, and cooperation in the introduction, implementation, and application of technologies. Technical cooperation consists of two main implementation methods: technical cooperation projects in recipient countries and projects that accept participants into training in Japan.

Japan International Cooperation Agency (referred to as JICA) is an ODA (Official Development Assistance) implementing agency that was established in 1974 as a special public corporation under the jurisdiction of the Ministry of Foreign Affairs by merging the Overseas Technology Agency, the Overseas Immigration Corporation, the Overseas Agricultural Development Foundation, and the Overseas Trade Development Association. JICA implements technical cooperation projects for international assistance, and the programs and their targets are set according to the achievements and progress of maintenance and management in recipient countries.

The authors have drawn on decades of experience, including ODA and domestic projects in Japan through industry–academia collaboration. The following are summaries of the authors' experience in transferring bridge maintenance management capabilities to Egypt and the Philippines.

In Egypt, bridge maintenance and management, mainly reactive maintenance or breakdown maintenance, known as a type of collective maintenance, have been conducted owing to the lack of inspection procedures. Against this background, the following activities were conducted as a technical cooperation project over three years [3]:

(1) Establishment of bridge inspection procedures, (2) inspection exercises, (3) classroom lectures on maintenance management methods, and (4) BMS (Bridge Management System) development and operational training.

On the other hand, in the Philippines, the following efforts were made to improve repair techniques as a new "Phase 3" after the development of inspection procedures and exercises on inspection and maintenance management techniques in Phases 1 and 2 for a total of six years [4].

(1) Revision of repair manuals, (2) onsite training (improvement of repair techniques), and (3) establishment of a database for repair history and implementation of operational training. Efforts of the project in the Philippines differ from those in Egypt in terms of the progress of technology transfer; they implemented sustainable projects that their concerned organizations have actively undertaken.

Both projects met the prescribed requirements and were completed. However, some points can be further improved for better technology transfer on an ongoing basis in terms of sustainability. The following example provides an impetus for the improvement methodology discovered through the authors' practice.

Martinez [5] pointed out the following problems with ODA in Africa, which were carried out by the European Commission and World Bank. African government initiatives were established during the colonial period, and they have a form of organization that operates for developed countries that do not fit in the African lands. This alienates African culture and is not connected to social realities despite decades of operation. In Africa, tribalism has led to high-ranking officials having a great deal of power, which has prevented the proper operation of the above government initiative. Planning is a Western concept that is complex, difficult, and impractical. Achieving efficient planning by providing foreign technical assistance formats to the Ministry of Public Works without modification is difficult.

La Chimia et al. [6] referred to the 2010 OECD report [7] and discussed the risks to donors following their own system and the importance of ensuring sustainability by utilizing the recipient country's system. Effective planning and operation require considering sustainability over several generations.

This issue in Africa was emphasized in 2001, yet there are many examples of ODA creating the same problem today. This can be understood from the fact that the World Bank has many problems and requires a review system such as the Inspection Panel [8].

In the late 1980s and the early 1990s, safeguarding policies governing environmental and social considerations in lending were stricter than the actual practices of most borrowing governments, a situation ignored by bank staff as the project progressed. Therefore, the Inspection Panel was established as a way for citizens of countries in which bank-financed projects were being implemented who were adversely affected environmentally and socially to request a review of the bank's compliance with its internal rules [9]. Lukas noted that the Inspection Panel, resolved in 1993, demonstrated that the underlying concepts are still valid today [10].

The conflict with sustainable development can be inferred from the above African and World Bank cases. In the African ODA case, there were cases of incompatibility in trying to follow the government system as it was in the developed country. In the case of the World Bank, there were cases where protection policies were too strict, and the procedures were not followed.

These also apply to the Japanese ODA projects that this paper studies as examples. The policy is to export superior Japanese technology and to downplay any nonconformities identified during the project when it places more emphasis on progress. Since these situations are also exposed in Japanese technical cooperation projects, the authors will describe them in this paper based on their experiences.

It is vital to define human, information, and organizational capital as intangibles formed in technical cooperation projects. According to Kaplan and Norton, it is necessary to focus on creating sustainable value in the future as a project strategy [11]. In the JICA projects, a monitoring and evaluation cycle based on the PCM (Project Cycle Management) method was used to correct the course of project implementation. For this monitoring and evaluation, evaluation items based on the DAC (Development Assistance Committee) CRITERIA [12] were used.

However, Yokokura et al. demonstrated that the five DAC evaluation items—RELEVANCE, EFFECTIVENESS, EFFICIENCY, IMPACT, and SUSTAINABILITY—did not evaluate the technical contribution of the project [13]. In addition, because the technical contribution is intangible, Shimizu pointed out that there is a risk that many dimensions will be ignored when measuring other quantitative indicators, while the evaluation of the technical contribution remains ambiguous; the phenomenon will only be captured by those indicators [14]. Shimizu also demonstrated that "efficiency" in a project indicates the ratio of calculation to input resources. In many cases, the minimization of human and physical resources or the time invested in meeting the required specifications may result in cases where sufficient effectiveness cannot be demonstrated [15].

The lack of proper project outcome definitions and evaluation items as intangible assets has caused the inclusion of uncertainties, which inhibits the effective action of trajectory modification.

These highlight the risk of placing too much emphasis on project efficiency when proceeding with project closure as the primary goal, even when there are indications of the need for additional resource input or an extension of the project's duration.

This study identifies and discusses these issues based on technical cooperation projects that the authors have experienced, as well as the good practices for improving maintenance management capabilities in Japan. First, the status of the projects in which the authors were engaged is reviewed from the perspective of the aforementioned issues. Next, the issues and improvement methods for each project task are discussed. Finally, we explore effective management methods for future technical cooperation projects.

## 2. Subject Matter and Issues

### 2.1. Target Projects

This study examines technical cooperation projects in international cooperation. The focus was on the organizational and individual engineering levels. At the organizational level, this study examined the goals and strategies for fostering "effective ownership" and the PDCA (Plan-Do-Check-Action) cycle, including measuring effectiveness. At the

individual engineer level, we examined the need to accumulate and transfer experience as a method for establishing a sustainable management cycle.

### 2.2. Issues to Be Addressed

In Japan's technical cooperation, monitoring is conducted using Project Cycle Management (PCM), which is a method for managing the entire cycle of development from formulation and implementation to evaluation. This method uses a summary table called the Project Design Matrix (PDM) that shows the logical inter-relationship among the project components to manage the project. The flow of monitoring begins with an understanding of the project's current status to identify and plan for problems and conditions. Next, the causes and impediments are identified through analysis. Finally, a cycle of improvements is determined to formulate countermeasures and new plans.

Figure 1 shows an overview of the strategy pyramid and the PDM. In the strategy pyramid, the mission and objectives are defined according to the goals established for the project, necessary strategies are formulated, and the project is defined accordingly. The PDM defines specific project goals and outcomes, as well as the activities and resources required to achieve them. These are not fixed, and the PDM must be rescheduled as necessary during the project's planning, implementation, and evaluation phases. Practically, however, there are many situations in which problems are overlooked because of the excessive emphasis on project progress and quantitative evaluation defined at the time of planning.

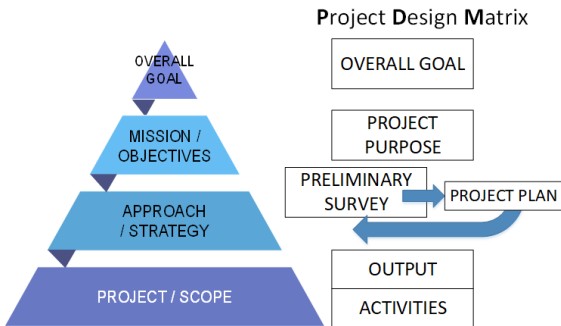

**Figure 1.** Strategy pyramid and PDM structure.

Furthermore, it is essential to materialize a strategy by conducting a "detailed planning study" before starting a project. However, in many cases, the project definition is fixed as that assumed during the initial project formation stage.

Figure 2 shows how the PDM can be used to determine the cause of a problem and improve the cycle. When monitoring results indicate that project goals are not being achieved as planned, the cause may be the lack of results or external conditions inhibiting the achievement of the goals. If a problem exists with the results, it can be inferred from the PDM that the activities and external conditions next to them cause the problem. Similarly, if there is a delay in an activity, the cause is searched from the inputs and preconditions. If there is no cause, even after investigating all possible causes based on the PDM, it is necessary to consider that the plan itself has a problem [16].

Thus, although flexibility is originally required without being fixed on the project definition assumed at the initial stage of project formation, the situation in which appropriate monitoring and correction do not work is discussed from a project experienced by the authors.

Figure 3 illustrates the three issues in technical cooperation experienced by the authors. The following sections discuss these issues in detail and explore areas for future improvement.

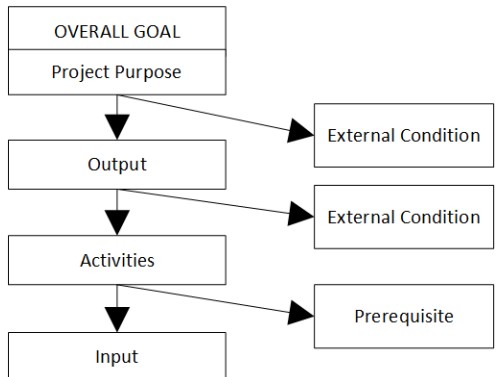

**Figure 2.** Causal analysis.

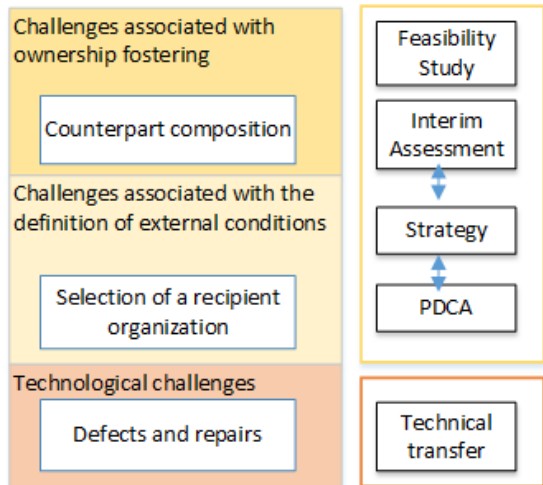

**Figure 3.** Issues in technical cooperation.

Issue 1: Issues related to methods for fostering ownership

It is desirable that the cycle of knowledge and experience gained through technical cooperation be continued in the recipient country even after the project has been completed. This section discusses issues that arise in the "composition of counterparts", which is the key to this process.

Issue 2: Definition of external conditions

The approach to overseas technical assistance begins with a "detailed planning survey" in the initial stage of the project formulation. In this process, potential impediments are defined as external conditions; however, for the recipient country, they are internal conditions [17]. An organization's structure and maturity are important issues.

Issue 3: Technical issues

Initial failures and repair cases were identified as issues during the implementation of the technical cooperation in the recipient country.

## 3. Actual Issues

### 3.1. Issues Related to Methods of Fostering Ownership

In Egypt, problems related to the composition of counterparts have emerged. This is illustrated in Figure 4. The team composition for the management of the BMS requires the cooperation of engineers in the maintenance bureau. However, because of a lack of personnel, several primary assignments were made from other bureaus, and a request was made to "train these staff members". We complied and proceeded with the task symptomatically because of the need to achieve results in a short period with limited resources.

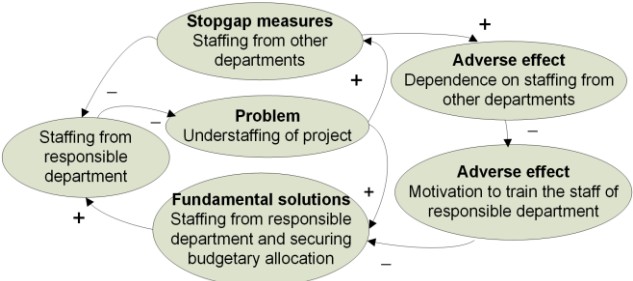

**Figure 4.** System model for Egypt.

Consequently, at the end of the critical phase, our counterparts informed us and the team that the team would need to focus on its core business and would be disbanded. This is an example of a situation in which the problem cannot be resolved without taking on a fundamental solution.

A system prototype [14] is a system model that typifies the problems observed in terms of the behavior of social systems. This model defines the adverse effects caused by emergency measures when such a problem occurs as "problem shifting". It is necessary to verify the situation in each phase of the project using such models to check whether "the root of the problem has been solved" or if "side effects have not occurred owing to symptomatic treatment".

In this case, there is a deep-seated problem with the recipient country's perception of local efforts towards technical cooperation as a special event during the project period. Efforts and evaluations from the perspective of ownership recognition are necessary.

### 3.2. Issues Related to Defining External Conditions

Figure 5 shows the organizational structure of infrastructure construction and maintenance in Egypt. First, upstream areas, such as accounting standards, are under the jurisdiction of the Ministry of Housing, not the General Authority for Road, Bridge, and Land Transport (referred to as GARBLT). Initially, the Ministry of Housing (Ministry of Housing, Utility, and Urban Affairs: HUUC) controlled Egypt's civil engineering field. The GARBLT has a history of separating its road division from HUUC. Currently, design standards are not harmonized as they are established by the Ministry of Transportation, which is the superior ministry of the GARBLT.

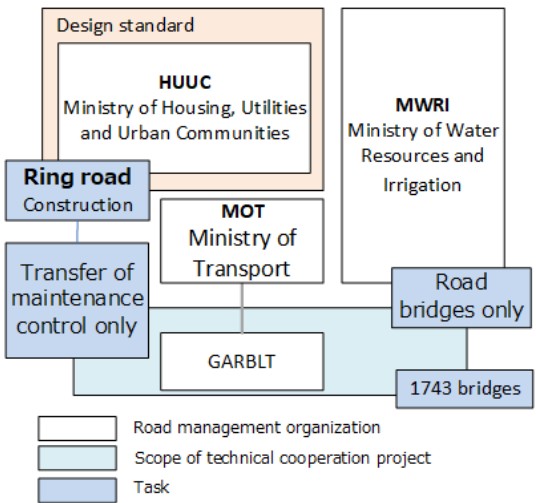

**Figure 5.** Road management organization structure in Egypt.

The ring road, the main urban road in Egypt, was constructed by the Ministry of Housing and transferred to the GARBLT, which maintains it.

The Ministry of Water Resources and Irrigation (MWRI) constructs and manages important river bridges, and the GARBLT manages only the roads over these bridges. In many countries, ports, roads, and railroads are differentiated after the initial creation of a ministry that controls all infrastructure. Similar to the problem of the counterpart team that operates BMS, this is a situation where the issue of "problem shifting" regularly arises because the recipients of technology transfer are essentially multiple organizations. In the case of the project in Egypt, the technology transfer was only to the GARBLT. This indicates the importance of selecting counterparties at the "detailed planning study" stage as a fundamental improvement of the problem.

Also, in the Philippines project, the network used by the Department of Public Works and Highways staff was not thoroughly investigated during the detailed planning study. At the beginning of the project, it was assumed that the staff would have an adequate network environment in all government buildings. However, it turned out that the WAN network contract was for 1 GB of bandwidth at the central office but only 10 MB of bandwidth at the regional offices. The problem made it impossible to fully utilize the network for developing a database system for repair histories.

The following three issues are pointed out in "Problems with PCM methods in development assistance" [15].

- The process of bringing the PDM to completion at a workshop was undertaken because of the organizational management's desire to quickly finalize the project, leading to poor preliminary investigation work and problems with the main body of work.
- The first full-scale PDM verification and feedback were conducted during the interim evaluation survey, and it was difficult to return to the aforementioned scoping issues when this stage was reached.
- Quantitative indicators are desirable in the PCM method. While this is expected in a management method developed as a type of MBO (management by objectives), measuring a project's results with quantitative indicators ignores various other dimensions, and the project is captured only by those indicators. There is a need to use quantitative indices when considering the dangers of mechanistic approaches.

To address these, the scope definition must be adjusted in terms of identifying counterparts and selecting specific technical cooperation targets. In this case, it is desirable that HUUC and MWRI be selected as targets of technology transfer and consultation and that the appeal be made in upstream areas, such as design.

Next, the following two examples will show why the preliminary investigation was not conducted sufficiently. Matsumoto [18] points out that in World Bank projects of international development finance institutions, "the relationship as a donor is reversed in the process of the survey, and there is a current situation in which they accept surveys that are not under their policies in consideration of the borrower". As a specific example, he cited the case of the Bujagali Hydroelectric Power Project in Uganda, where a preliminary survey was not conducted, although land speculation that occurs with the development plan and population inflow associated with the project were predicted in advance. The report cites a case in which the actual situation was not identified because of concerns about economic losses caused by delayed decision-making by the government of the borrowing country and the project proponent.

According to a 2006 evaluation report published by the Asian Development Bank [19], a preliminary study on the number of residents affected by evictions, land use, and assets in development projects financed by the bank showed that the percentage of relocated residents was 10% higher than the initial estimate and that the number of residents who were not relocated but affected by the land loss was 1.5 times higher than the initial estimate. In this case, the study points out that the problem was not in the survey methodology; the survey was conducted with the expectation that it would affirm development.

Thus, there are cases involving similar issues in a single country as well as in international development finance institutions, which are financing projects in multiple countries. It is necessary to share cases and study countermeasures beyond Japan's aid framework.

### 3.3. Initial Defects and Technical Deficiencies in Repairs

Inspection with counterparts revealed that many defects were caused by design flaws, construction problems, and age-related deterioration owing to heavy traffic and other loads. In addition, many repair cases did not fully consider the roles of the functional members. However, because this is a problem that should be handled with care and attention to the relationships among the sections of the road management organization in the country and with the external organizations mentioned earlier, we did not propose remedial measures during the project implementation period. The following is a list of the design, construction, and repair problems in the case of the project in Egypt:

(1)    Construction conditions around the bearing (design and construction problems)

Concrete and reinforcing steel bars were placed around the bearing to prevent it from moving (Figure 6).

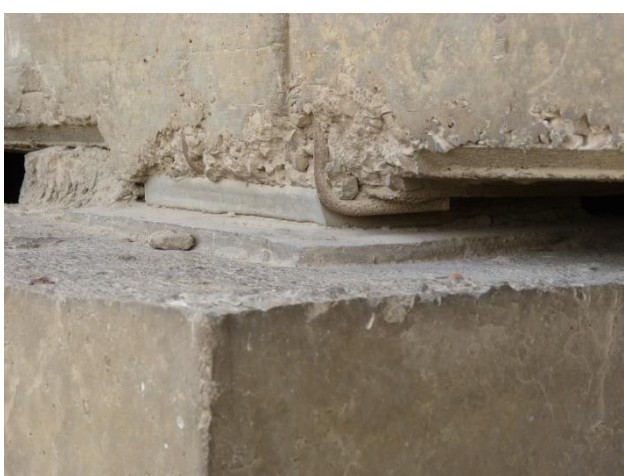

**Figure 6.** Inadequately constructed bearing zone.

(2)    Girder damage owing to insufficient building height limit (design planning problem)

The heights of vehicles passing under the girder (building height limit) were not considered. The possibility of bridge failure is high because of the insufficient load-bearing capacity caused by the breakage of the steel bars in the girder (Figure 7).

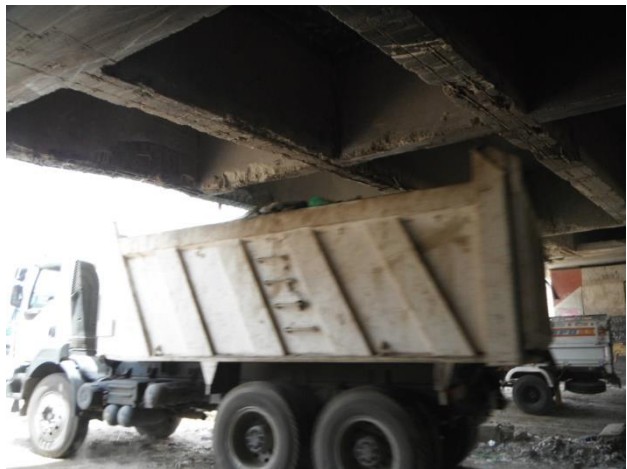

**Figure 7.** Girder damage caused by inadequate vertical clearance.

(3)    Repair of expansion device using high viscoelastic binder (erroneous repair)

When the comb-shaped expansion devices need to be replaced, owing to dependence on overseas imports, the parts near the expansion and contraction are considered small and are replaced by a unique pavement construction on the joints using a highly viscoelastic binder and aggregate (Figure 8).

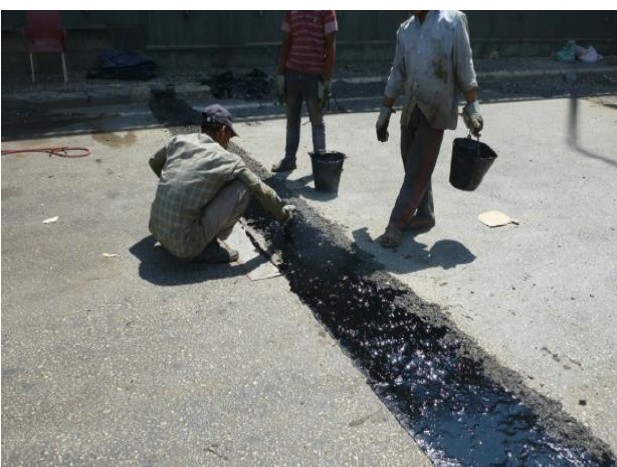

**Figure 8.** Repair of expansion joints using highly viscoelastic binder.

This means that although it was thought at the design stage that steel finger joints are necessary because of relatively large amounts of expansion and lateral clearance as a safety margin, on-site judgments are made according to an erroneous economic principle. Consequently, "repair" to maintain the initial performance and functions of the structure is not conducted, and the structure becomes undetectable for abnormalities in the expansion space owing to damage.

In such cases, where the recipient country outsources construction as well as inspection, check-up, and diagnosis, there may be insufficient progress in improving the capabilities of in-house engineers.

If the capability to infer the inspection/diagnosis results submitted by the external organization and judge the validity of the inspections conducted by the external organization at the time of completion are insufficient, the structure will have the problems described above.

The following two approaches can be considered for technical cooperation.

(a) Refer to inspection systems and standards to strengthen management in upstream processes, such as design.
(b) Bottom-up approaches to strengthen judging ability and raise awareness for future new construction and rehabilitation by using existing bridges as a teacher to gain "awareness".

Regarding (a), as described in "Selection of target organizations for support" in the Issue mentioned above: Definition of external conditions. It was difficult to appeal to organizations with jurisdiction over design standards, as they were not targeted for technology transfer. The approach in (b) is described in Section 4.

### 3.4. Optimization of Life Cycle Cost

After focusing on construction to alleviate quantitative shortfalls, road management organizations are faced with deciding whether to extend the service life of bridges by repairing components or removing old bridges and constructing new ones; reconstruction as a large-scale maintenance management involves the complex problem of how far to consider the optimization of Life Cycle Cost (LCC) from a long-term perspective. Since there are no examples of overseas projects, this paper introduces a characteristic case in Japan.

## 4. Practice and Solutions

Civil engineering is often considered experiential engineering. Accordingly, the three issues defined in Section 2 which present mechanisms for thinking and practice to provide counterpart engineer a sense of project activities as output-type learning.

### 4.1. A Practice for Fostering Ownership

Figure 9 shows an example of a subject, "project exercise", of the "*MICHIMORI*" training program in Nagasaki, Japan, which creates a review forum and elicits improvements in technology and awareness. *MICHIMORI* is a program for human resource development aimed at acquiring knowledge and skills related to the maintenance and management of infrastructure facilities. It has been in continuous operation since 2008 [20].

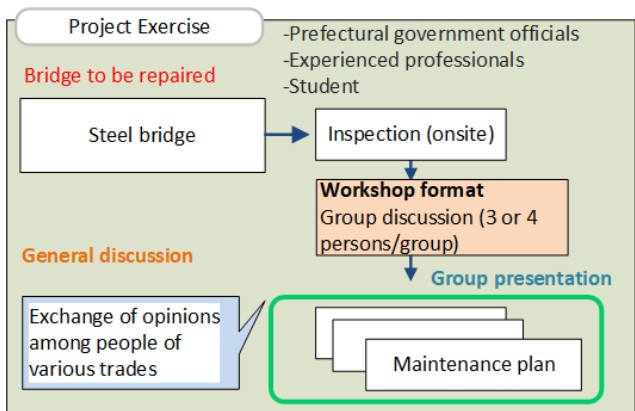

**Figure 9.** Project exercise in Nagasaki prefecture.

In the "project exercise" of the program, trainees of the program, such as Nagasaki prefectural government officials, municipal government officials, and experienced workers of the private sector in the region, held discussions and formulated a management plan for each bridge. This was a valuable opportunity for people from various industries, such as construction, design, and local governments, to exchange opinions and learn freely. Every year, a questionnaire is sent to all participants for analysis; one participant commented, "I gained confidence in thinking, deciding, and explaining by myself". Furthermore, they courageously argued for and corrected the inappropriate diagnostic proposals.

It is practical to create a system that allows engineers to ask questions and prioritize them to make maintenance and management more effective.

### 4.2. Solutions to the Definition of External Conditions

(1)   Reconsidering how to proceed with pre-survey work

It is practical to reconsider the role of the executing agency and the method of implementation and compilation of the consultants commissioned to conduct the study.

(2)   Reconsidering the conduction of interim evaluations of each project

Currently, the project is being coordinated by a contracted consultant; therefore, there is a tendency to conclude that the project is progressing as planned. It is practical to change the policy and create a "system" to enable prompt response to corrections by breaking away from the thinking based on economic principles on the part of the contractor for individual contracted work. By relieving the contractor from the pre-survey, which focuses on time and cost, it is possible for the contractor to intervene deeply in the recipient country's maintenance management system and situation.

(3)   Reviewing the evaluation axes for PDM goals, outcomes, activities, and inputs in the PCM methodology.

The German GTZ (Deutsche Gesellschaft fur Technische Zusammenarbeit) developed the PCM method based on a logical framework devised in the United States. The German GTZ, an international aid organization similar to JICA, has been using this method for planning assistance in all its technical cooperation projects since 1983. The method was also adopted by many other aid-implementing agencies, including NORAD (Norwegian Agency for Development Cooperation).

Currently, in the German GTZ, the PCM methodology does not have a mandatory requirement but must be used only in specific cases [14]. As mentioned in Section 3, measuring the project results with only quantitative indicators results in the neglect of various other dimensions; the project is captured only by those indicators. The danger of a mechanistic approach [14] is related to this problem, and an evaluation model for the PCM method must be reviewed.

### 4.3. Solutions to Technical Issues

(1) Examples of domestic efforts concerning initial failures: test construction

In Japan, many cases were equivalent to initial failures like shoddy workmanship during the Showa period (1926 to 1989). To date, there are many problems caused by the quality of concrete, such as "cracks owing to drying shrinkage" and "bean slabs"; improving the quality of concrete remains an issue to be addressed in the future.

This paper introduces the efforts made in Iwate Prefecture, Japan, which experienced the Great East Japan Earthquake, with an emphasis on empirical engineering [21].

To ensure the quality of the concrete, a test construction was conducted on a reconstructed road, and a vibrator was applied on a 40 cm square area for 6 s and removed for 2 s (Figure 10). In addition, the curing period was set to be longer than typical (at one month) to maintain the wet condition of the water conveyance sheet.

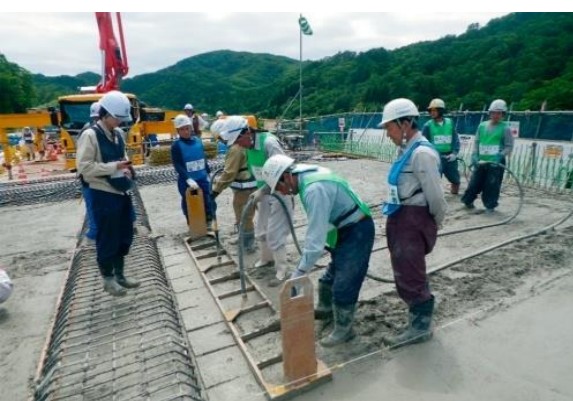

**Figure 10.** Test construction (vibrator compaction).

Based on the advice of academic experts, they repeated the test with a contractor to achieve zero cracks of 0.2 mm or more in width.

Although gathering on-site engineers for test construction was challenging owing to the COVID-19 pandemic, this is a good example of a careful "root solution aimed at the root cause of a problem". Through practical work, such as output-type learning, this method can be expected to solidify knowledge about error prevention.

Thus, it is practical to educate counterparts through experience by carefully dealing with them when they encounter initial defects in overseas projects.

(2) Discussion mechanism: damage assessment review

Figure 11 shows a counterpart-led damage assessment review conducted in Egypt. The authors were inspired by the Osaka Prefectural government's service life extension and repair plan, in which the staff in charge of bridges at each civil engineering office gather at the main office to present the damage status of bridges under their jurisdiction

and share their inspection results and analysis. Consultants were commissioned to support the planning process and commented on the validity of the ratings and factors that led to damage in the actual damage example.

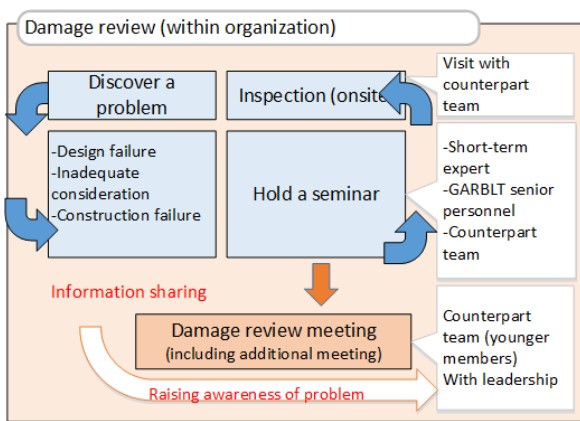

**Figure 11.** Damage review meeting.

Figure 12 illustrates the discussion focusing on the cases of initial failure. Discussing the causes of these defects is essential as an effort to create a system in which all participants can share their experiences.

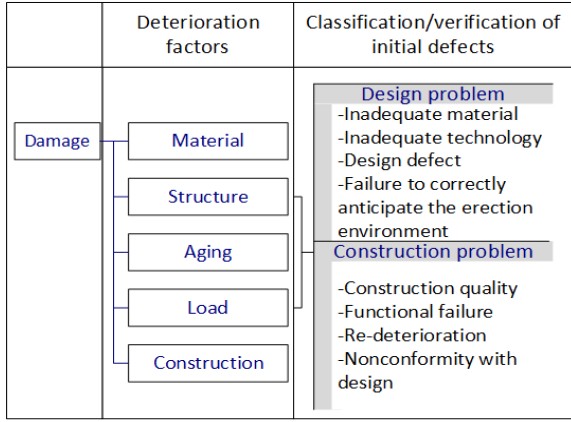

**Figure 12.** Organizing ideas for consideration and discussion of deterioration factors.

(3) Acquisition of elemental technology: eye-tracking system

Inspection and repair manuals have been prepared for both Egypt and the Philippines, and field training was conducted; however, we believe that this is insufficient. Therefore, the following is a proposal for improving the effectiveness of "experiential engineering".

Eye-tracking systems have been used in various cases and are now being used effectively in Japan to reduce the efforts of skilled workers in factories. In civil engineering, Eguchi et al. [22] used this system to improve bridge inspection accuracy.

The results of the soundness diagnosis of the rebar exposure and corrosion of a concrete bridge were analyzed, and a difference in the judgment values was observed between a skilled engineer and an engineer with five years of experience. Concrete bridges, even for bridges of the same type and scale, have significant material and construction issues that affect their performance in terms of long-term use. In this case, the damage process of poor-quality concrete was determined to be in the neutralization stage, which was slower than the corrosion process caused by salt damage. In addition to visual judgment, experience and knowledge cultivated over many years are important for assessing damage through inspection, and it is necessary to pass on this knowledge.

"Empirical engineering" is often associated with years of experience and intuition. However, the results of the studies mentioned above are "engineering judgments" based on civil engineering technology. Even if the technology is not handed down comprehensively for all bridge types and cases, it is an "important experience" to be exposed to the knowledge and causal relationships through such analysis of actual damage cases in the recipient country. This enables trained engineers to impart the gained knowledge to young engineers after completing a technical cooperation project.

(4) Example of reconstruction considering Life Cycle Cost

The Atsuta Denma Bridge was built in 1950, and 65 years have passed since its construction, and it has deteriorated.

It is the oldest of the 64 overpass bridges built after the war on National Route 1, managed by the Ministry of Land, Infrastructure, Transport and Tourism.

Generally, extending the lifespan of a bridge by repairing or replacing components will reduce costs, but this is a good practice example of rebuilding a bridge considering LCC. Figure 13 shows a drawing of the Atsuta Denma Bridge [23].

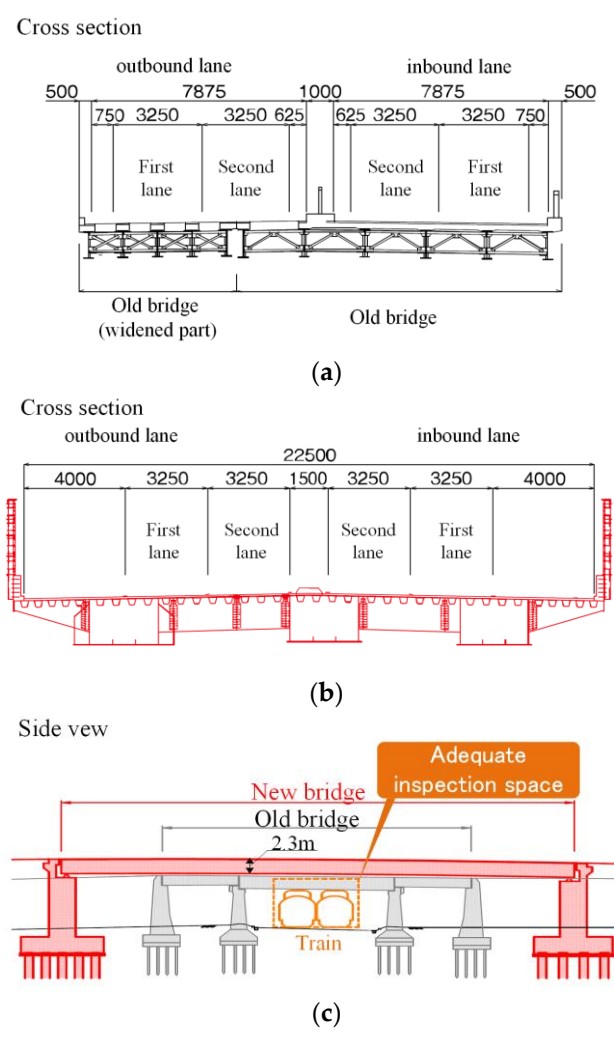

**Figure 13.** Replace of Atsuta Denma Bridge (unit: mm): (**a**) Old bridges; (**b**) New bridge; (**c**) Clearance for maintenance.

- The old bridge had a high maintenance cost because of the method of adding a 1.5-lane bridge and connecting it with deck slabs when it was widened.
- The new bridge is a box-girder structure with a sufficient inspection corridor to facilitate internal inspections.

- The old bridge was a railroad crossing, and the distance between the old bridge and the railroad tracks was too narrow to allow inspection by workers at height, a condition that further increased the cost of the bridge.
- The old bridge was a riveted bridge, and repairs had become difficult due to a decrease in the number of "riveter" craftsmen.
- The bridge was designed to meet the latest earthquake resistance standards to minimize damage in the event of an earthquake.

This case study has several implications for bridge maintenance management.

First, as mentioned at the beginning of this paper, the replacement of the bridge was implemented not only by comparing construction costs but also by considering future maintenance, which is usually challenging to determine due to many uncertain factors.

Secondly, not only is the bridge being replaced, but the structural type and painting specifications are also being considered to minimize the LCC actively. In developing countries, constructing new bridges is still a significant means of alleviating the quantitative shortage of infrastructure under the demographic dividend. However, as society matures, it is essential to consider construction and maintenance methods that aim to minimize future maintenance and management costs, as in this case in Japan.

## 5. Discussion

### 5.1. Examining the Effectiveness of Each Solution

This chapter examines the non-case-specific effectiveness of the solutions described earlier and describes important considerations. Expected effects are discussed separately for efforts to strengthen management at upstream processes and bottom-up efforts associated with existing bridges and bridge sites. Figure 14 shows scopes of different types of efforts.

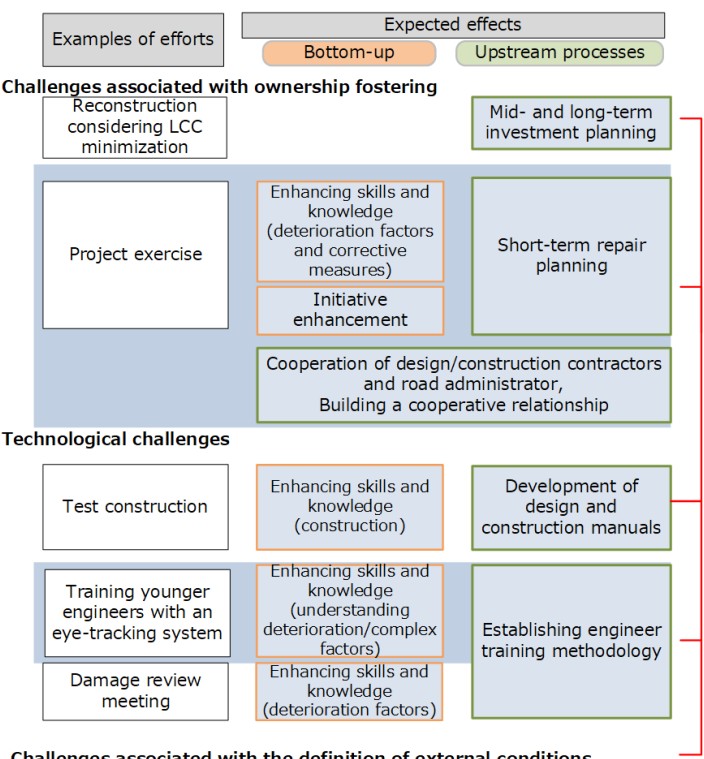

**Figure 14.** Scopes of different types of efforts.

Of the three challenges mentioned earlier, the challenges associated with the definition of external conditions are not indicated as targets. As shown with red lines, however, they are related to other challenges. This relationship is mentioned in the "important considerations" in each case.

(1) Example of reconstruction considering LCC

In the case of BMS developed by authors for domestic and overseas, the mid-to-long-term investment plan is a simple one that provides a solution for replacing a bridge when its soundness exceeds an acceptable limit in predicting deterioration based on numerical calculations. On the other hand, the case of reconstruction presented in Chapter 4 is a good practice of LCC consideration that includes uncertainties but considers future maintenance costs. This approach will be an important factor in considering future mid- to long-term maintenance plans.

(2) Project exercise

From a bottom-up approach, project exercise can be expected to enable participants to acquire skills and knowledge outside their specialty areas because people in different fields, including prefectural government personnel, practitioners, and trainees, qualified as MICHIMORI participants.

From the viewpoint of ownership fostering, project exercise can be expected to motivate participants to take the initiative in maintenance through group presentations and discussions on maintenance planning. With respect to effects related to upstream processes, project exercise is thought to be an excellent form of output-type learning because the ability to draw up short-term repair plans can be acquired through discussions on bridge management planning.

One important consideration is that, as mentioned earlier, the involvement of local contractors and consultants should be considered carefully. Flexible interpretation and exchange of opinion may be hampered because they are hierarchically structured instead of being divided into design and construction groups.

Therefore, in cases where different contractors are assigned to different areas, devising a system for enabling the participation of other contractors (e.g., cross-zoning) so that contracting is not restricted by interests may be effective. This is related to the challenges associated with the definition of external conditions.

From the point of view of neutrality, it may be necessary to assign academic experts as moderators to encourage the participating designers, contractors, and road administrators to express flexible opinions.

(3) Test construction (addressing initial defects)

From the point of view of a bottom-up approach, in comparison with input-type learning using manuals for new construction and repair projects, test construction is highly effective as output-type learning involving hands-on activities. With regard to upstream processes, it may be possible to develop manuals and standards from a new point of view, such as including descriptions or provisions associated with efforts to prevent construction defects, a subject that is usually touched on in such documents.

An important consideration is that, as mentioned in connection with the example of the repair of expansion joints, there is a tendency to carry out new construction or repair in accordance with an erroneous concept of economic efficiency. The authors think, therefore, that, as mentioned in connection with the example, a form of assistance by a team that includes domestic construction company members and academic experts is effective. Such assistance makes it possible to call attention to problems related to not only the construction work itself but also upstream process details such as inspection systems. Local contractors that experience test construction may also be involved in construction or repair projects in the areas (subdivisions of administrative districts) to which the contractors are assigned. It is, therefore, necessary to select contractors and use appropriate exercise processes that take geographic coverages into consideration instead of selecting model regions and carrying out test construction only in a small number of areas. This is related to the challenges associated with the definition of external conditions.

(4) Training young engineers with an eye-tracking system

From the point of view of a bottom-up approach, this method is related to the wisdom of experts mentioned in Chapter 3. This method aims to enable engineers to respond to different events flexibly (What is the understanding of an expert?), hypothesize deterioration mechanisms, come up with inspection and diagnosis approaches, and hand down their knowledge to younger engineers. This approach, therefore, contributes to skill and knowledge enhancement by helping people gain a deeper understanding of deterioration and complex factors. With respect to effects on upstream processes, this approach is thought to contribute to developing new engineer training curricula based on the analysis of such knowledge. Hence, the recommended approach is to pay attention to the fact that the effectiveness of this method can be enhanced significantly through engineering judgment, which aims to make sound judgment based on civil engineering when evaluating structural integrity in order to analyze thought processes.

An example of a successful attempt at deep learning has been reported (Hyodo et al. [24]). In the reported case, interview surveys were conducted with experienced engineers who practice engineering judgment instead of relying entirely on data, and experts' domain knowledge was extracted. The domain knowledge thus extracted was broken down into many attributes that can be readily understood by non-experts, and a flow of procedures for presenting the rationale for the judgment was successfully established.

In the precision teaching method in behavioral engineering (Lindsley [25]), it has been pointed out that complex thoughts can be thought of as a combination of simple evaluations. Thought processes associated with structural integrity diagnosis can also be broken down into smaller parts and can be thought of ultimately as a combination of assessments. The authors believe that if important data can be shown by analyzing engineering judgments and paying attention to the relationships among damage, deterioration phenomena, and agents, it is possible to acquire the knowledge needed to answer the question "What do experienced engineers evaluate?"

With respect to system implementation, it is necessary to sufficiently fund the project because activities such as device utilization and system-based analytical processing need to be performed so that the recipient country can make effective use of the acquired knowledge and techniques.

(5) Damage review meeting

From a bottom-up approach, damage review meetings can be expected to help enhance skills and knowledge because road administrators and consultants jointly discuss the causes of damage. Regarding effects on upstream processes, damage review meetings are thought to contribute to developing new engineer training curricula, in view of the experience gained by letting young engineers in Egypt chair and participate in those conferences.

An important consideration is that the participation of local consultants is hoped for on a sustainable basis instead of only during the technical cooperation project. In the case of the project in Egypt, if a bridge shows a sign of damage that requires early repair, the consultant hired by the construction contractor is to propose an inspection/investigation/remediation approach.

Such contractual relationships are likely to hamper the flexible exchange of opinions.

There is a need to come up with a way to reduce the influence of interests. We should explore ways to establish or restructure industry-wide systems, taking account of upstream processes instead of trying only to promote the acquisition of element technologies.

This is related to the challenges associated with the definition of external conditions. During the project period, it is necessary to visit the region where the project site is located as much as possible without caring only about accessibility. Introducing elements of trainer training to give positive effects on regions other than the model region is thought to be effective.

(6)    Considering solutions to external conditions

The review of the interim evaluation method and the review of the evaluation axis of PDM described in Chapter 4 can be made possible by deeply intervening in the formation of ODA projects. Currently, private consultants participate in projects after the work is coordinated. Therefore, it is necessary to consider and propose such higher-level perspectives individually as the other solutions described here are carried out.

*5.2. Consideration of Suitability for PDCA in Asset Management*

Figure 15 shows a multi-tiered cycle in asset management. The solutions described in this chapter and the contributions to the bottom-up and upstream perspectives are adequate for each PDCA cycle in this figure.

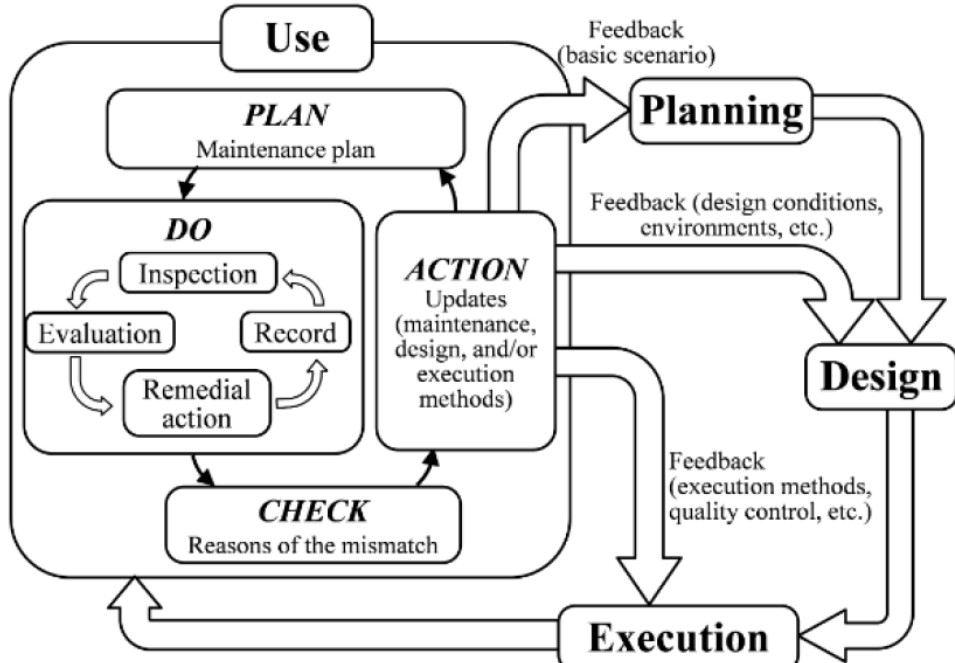

**Figure 15.** Maintenance management cycle. Reproduced with permission from Yokota, H.; Nagai, K.; Sakai, K. 2022. [26].

The test construction and the eye-tracking system for young engineers contribute to the innermost cycle. Mismatches in this cycle are picked up as opportunities to reconsider the external conditions shown in Figure 14.

Project exercises and damage reviews effectively contribute to the middle PDCA cycle and the topmost cycle. The feedback of the PDCA cycle in the middle level is effectively provided by fostering opportunities for engineers in multiple positions to participate and exchange opinions in the project exercises.

## 6. Conclusions towards Future Discussions

This study extracted issues from technical cooperation projects conducted by the authors and examined improvement methods for each problem. Effective improvement methods were sought from overseas and domestic projects in Japan.

The three primary issues and the countermeasures and effects to address them are listed in Table 1. The following were identified as issues that could be improved: The project evaluation in JICA is not focused on the issues mentioned but on the results of activities that are easy to understand, such as "preparation of manuals, implementation of on-site training, and introduction of the maintenance management system".

**Table 1.** Issues and countermeasures.

| Issues/Measures | Title | Effect |
|---|---|---|
| Issue 1 | Challenges associated with ownership fostering | |
| Countermeasure | Project exercise | To be an excellent form of output-type learning [5. Discussion (2)] |
| Issue 2 | Challenges associated with the definition of external conditions | |
| Countermeasures | Revision of detailed planning survey procedures Improving the mid-term evaluation method Review of the evaluation axis of the PCM method | There is little direct influence [5. Discussion (6)] It will become more feasible in the future as consultants can participate in projects at an early stage [6 Conclusion towards future discussion] |
| Issue 3 | Technological challenges | |
| Countermeasures | Test construction | Highly effective as output-type learning involving hands-on activities [5. Discussion (3)] |
| | Damage review meeting | Contribute to developing new engineer training curricula [5. Discussion (5)] |
| | Acquisition of elemental technology: eye-tracking system | Related to the wisdom of experts and the hand down of their knowledge to younger engineers [5. Discussion (4)] |
| | Reconstruction considering LCC | An important factor in considering future mid- to long-term maintenance plans [5. Discussion (1)] |

Issue 1 is related to the problem of focusing on project progress and responding symptomatically when a problem occurs.

Issue 2 is concerned with the challenges associated with the inappropriate definition of external conditions.

In addition to these two issues, the lack of assessment of the technical contributions of the evaluation method may prevent an appropriate improvement cycle from functioning.

The technical issues in Issue 3 require donors to question the effectiveness of their technologies and systems from the perspective of those providing technical cooperation.

The common point to be recognized in the above three issues is the risk of providing aid based on the institutions and mechanisms of the own country and the importance of ensuring sustainability by utilizing the institutions of the partner country.

Therefore, it is important to analyze, evaluate, and improve the current situation during each project, even if external conditions and preconditions are not sufficiently identified at the "detailed planning and survey" stage.

In addition, as discussed in Section 2.2, causal analysis, a flexible perspective is needed to include higher-level indicators of PDM as targets for improvement if there are problems with the project.

In the future, the following issues should be considered in continuous monitoring and improvement cycles.

- The period of review and evaluation for monitoring and improvement should not be time-bound but determined based on actual achievements at each stage.
- There should be room for elastically reconsidering indicators at higher levels without setting a deadline based solely on time limitations.

In December 2023, JICA introduced a system that allows Japanese private companies to be involved in Official Development Assistance (ODA) projects from the proposal stage. Until now, the Ministry of Foreign Affairs (MOFA) and JICA have coordinated the

projects with the partner emerging countries, and companies have only been involved after their participation in the project has been decided through subsequent bidding and other processes. The new system will allow companies to participate in the project from the project coordination stage and allow for more effective proposals to be developed with the participation of companies.

Issue 2, discussed in this chapter, will become more feasible when private companies can participate in the project at an early stage.

In response to this, the perspective of future research is to examine effective methods of implementing technical cooperation projects by following up on the efforts of private companies in higher-level studies, such as formulating strategies and PDMs, as described in Figure 1. In the future, we will participate in actual projects from the project formulation stage and compare and analyze the past project management and the future required project management.

**Author Contributions:** Writing—review & editing, T.M., T.N. and Y.K.; Supervision, T.N. and S.N. All authors have read and agreed to the published version of the manuscript.

**Funding:** This research received no external funding.

**Institutional Review Board Statement:** Not applicable.

**Informed Consent Statement:** Not applicable.

**Data Availability Statement:** The data presented in this study are available on request from the corresponding author.

**Conflicts of Interest:** Author Teruyuki Miyakawa was employed by Dia Nippon Engineering Consultants Co., Ltd. Other authors declare no conflict of interest.

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
