# Peer review of "Practice for Continuous Effect of Technology Transfer in Bridge Maintenance and Management"

_sustainability, doi:10.3390/su16020492_

Round 1

Reviewer 1 Report

Comments and Suggestions for Authors

The paper describes the international cooperation projects of the Japan International Cooperation Agency (JICA) for bridge maintenance and management in several countries. The authors share their experience and expertise identifying most important issues and ways of dealing with those in the future. The paper is well-structured, it is very interesting, and I have learned a lot while reading it.

There are some minor drawbacks that should be considered to improve the quality of the paper before publishing:

1) Double check section numbering. It is confusing with many sections named “Issues to be …” while being on different levels (Section 3, Section 2.2, Section 3.1).

2) Reposition figures to be closer to the place where they are mentioned. Currently you have to go back and forth sometimes.

3) There is no conclusion section. Section 5 contains some information that could be considered conclusion, but a separate brief conclusion section would increase the quality of the paper. Section 5 itself is more of a discussion of the results though.

4) Provide directions for future research or studies. That is briefly mentioned in the last paragraph of the paper with issues to be considered, but having a more specific goal for future research itself is crucial. That would make a better understanding of the current study’s place in the overall process.

5) Most references are from 2000s and 2010s with only three being from 2020s. Are there no other more recent references? If there are no recent references, then mention that in the introduction explaining why it is the case.

6) Please do not break words across a line-break by means of a hyphen. It is generally not advisable to do in scientific papers. If a word does not fit on a line, move it fully to the next line.

Once this is amended, the paper will be suitable for publication in this journal.

Author Response

Thank you for your review. Please refer to the attached word file.

Reviewer 2 Report

Comments and Suggestions for Authors

Article review for “Sustainability- 2740608”; “Practice for continuous effect of technology transfer in bridge maintenance and management”

The article starts with a fascinating subject, which could positively impact bridges’ functioning in the building industry and society. Durability is a broad subject covering the upper scale of the built environment to the construction, going down to the materials’ composition and performances. Strategies for planning and actions are of fundamental necessity for avoiding the risk of failure. Hence, in addition to Japan’s effort in supporting developing countries for infrastructure, particularly bridges’ efficiencies, the article explains the fundamental advantages of some methods and their application and priorities over each other. However, some essential components must be elaborated and modified to bring the research up to a publishable journal article level. Precise examples are provided in the following.

-             Content-wise, it is rather storytelling than solid research, hardly ever sharpening towards an actual robust study,

-             Structure-wise, the paper returns to the introductory statements on several occasions,

-             The study does not provide in-depth support for the abstract. The inconsistency is observable in instances such as failing to address the content of lines 19-21 on page one in the entire study.  

-             Strong statements need scientific support, such as literature, which is very often left unsupported (e.g., page 2, lines 46-54),

-             No experiments, no large variety of decently researched case studies, and no extended literature study (i.e., lines 519-568) are provided in the current state of the article,

-             There is no solid evidence that supports the research validity (e.g., page 10, section 5., first and second paragraphs, lines 486-488, and 489-493; page 11, table 1; etc.) and the outcomes explained in the article (e.g., on page 11, the storytelling style continues even in the end, providing expressive statements and summaries of the implication and connotation).

-             In general, the number of literature studied in this research is very limited (i.e., pages 11 and 12 contain only 21 references),  

Comments on the Quality of English Language

Only a few English errors were detected. 

Author Response

(The authors gave the same response as above.)

Reviewer 3 Report

Comments and Suggestions for Authors

The manuscript is clear and well written, easy to understand and profitable for the researchers in the field of infrastructure intergity and monitoring. Only a lack on economical aspects: the technical-economical balance is crucial to decide demolition/reconstruction instead of reinforcement. A typical case study is the following: Sassu, M., Puppio, M.L., Doveri, F., Ferrini, M., Mistretta, F., "A time and cost-effective strengthening of RC halfjoint bridges exposed to brittle failure: application to a case study" Structure and Infrastructure Engineering, 2023, article in press DOI: 10.1080/15732479.2023.2275689.

1. What is the main question addressed by the research?

A report on techniques to prepare maintenance reports on bridges, taking into account a large scale stock of infrastructures to manage, specifically referred to Japan

2. Do you consider the topic original or relevant in the field? Does it
address a specific gap in the field?

It is quite original, furnishing to the reader an interesting point of view, helpful for countries (as Italy) affected by maintenance problems on bridges
3. What does it add to the subject area compared with other published
material?

An interesting point of view to detect defects in large scale stocks   

4. What specific improvements should the authors consider regarding the methodology? What further controls should be considered?

The economical aspect is not taken into account: when you decide to demolish or reinforce a bridge is a crucial parameter, also in terms of life cycle and social damages   

5. Are the conclusions consistent with the evidence and arguments presented and do they address the main question posed?

Yes 

6. Are the references appropriate? Yes, but they can improved  

7. Please include any additional comments on the tables and figures.
No others to add

Author Response

(The authors gave the same response as above.)

Round 2

Reviewer 2 Report

Comments and Suggestions for Authors

No further comments!